# Genomic architecture and evolutionary antagonism drive allelic expression bias in the social supergene of red fire ants

Carlos Martinez-Ruiz[1]*, Rodrigo Pracana[1†], Eckart Stolle[1‡], Carolina Ivon Paris[2], Richard A Nichols[1], Yannick Wurm[1,3]*

[1]School of Biological and Chemical Sciences, Queen Mary University of London, London, United Kingdom; [2]Departamento Ecología, Genética y Evolución, Facultad de Ciencias Exactas y Naturales, Universidad de Buenos Aires, Intendente Güiraldes 2160, Ciudad Universitaria, Buenos Aires, Argentina; [3]Alan Turing Institute, London, United Kingdom

**Abstract** Supergene regions maintain alleles of multiple genes in tight linkage through suppressed recombination. Despite their importance in determining complex phenotypes, our empirical understanding of early supergene evolution is limited. Here we focus on the young 'social' supergene of fire ants, a powerful system for disentangling the effects of evolutionary antagonism and suppressed recombination. We hypothesize that gene degeneration and social antagonism shaped the evolution of the fire ant supergene, resulting in distinct patterns of gene expression. We test these ideas by identifying allelic differences between supergene variants, characterizing allelic expression across populations, castes and body parts, and contrasting allelic expression biases with differences in expression between social forms. We find strong signatures of gene degeneration and gene-specific dosage compensation. On this background, a small portion of the genes has the signature of adaptive responses to evolutionary antagonism between social forms.

**\*For correspondence:**
c.martinezruiz@qmul.ac.uk (CM-R);
y.wurm@qmul.ac.uk (YW)

**Present address:** [†]Department of Zoology, University of Oxford, Oxford, United Kingdom; [‡]Center for Molecular Biodiversity Research, Zoologisches Forschungsmuseum Alexander Koenig, Bonn, Germany

**Competing interests:** The authors declare that no competing interests exist.

## Introduction

Evolutionary antagonism can emerge when selection favors multiple phenotypic optima within a population. This process can lead to selection for reduced recombination between co-adapted alleles encoding the different phenotypes (*Charlesworth, 2016*). In turn, reduced recombination will sometimes favor the formation of supergene regions containing tightly linked alleles of up to hundreds of genes. Such regions enable the maintenance of genetic interactions over evolutionary time (*Darlington and Mather, 1949*; *Thompson and Jiggins, 2014*). We now know that supergenes controlling ecologically important traits are widespread. Examples include flower heterostyly in *Primula* (*Li et al., 2016*), mating type in *Microbotryum* fungi (*Branco et al., 2018*), Batesian mimicry in butterflies (*Joron et al., 2011*; *Kunte et al., 2014*), mating behavior in white-throated sparrows (*Zinzow-Kramer et al., 2015*; *Tuttle et al., 2016*) and male sexual morphs in ruff sandpipers (*Küpper et al., 2016*; *Lamichhaney et al., 2016*).

The non-recombining portions of sex chromosomes are extensively studied examples of supergenes (*Bergero and Charlesworth, 2009*). Their evolution is thought to have been shaped by antagonism over sexual phenotypes (*Zemp et al., 2016*; *Vicoso et al., 2013*; *Khil et al., 2004*; *Parsch and Ellegren, 2013*; *Wright et al., 2017*; *Mank, 2017*). However, studies focusing on young sex chromosome systems suggest otherwise (*Dufresnes et al., 2015*; *Nozawa et al., 2014*; *Alekseyenko et al., 2013*; *Muyle et al., 2012*; *Charlesworth et al., 2005*; *Stöck et al., 2011*). Indeed, sex chromosome evolution could be driven mostly by processes resulting from suppressed

**eLife digest** Red fire ants (*Solenopsis invicta*) are native to South America, but the species has spread to North America, Australia and New Zealand where it can be an invasive pest. A reason for this species' invasiveness types of colonies : one with a single egg-laying queen and another with several queens. However, it is not possible to simply add more queens to a colony with one queen. Instead, the number of queens in a colony is controlled genetically, by a chromosome known as the 'social chromosome'.

Like many other animals, red fire ants are diploid: their cells have two copies of each chromosome, which can carry two different versions of each gene. The social chromosome is no different, and it comes in two variants, SB and Sb. Each ant can therefore have either two SB chromosomes, leading to a colony with a single queen; or one SB chromosome and one Sb chromosome, leading to a colony with multiple queens. Ants with two copies of the Sb variant die when they are young, so the Sb version is inherited in a similar way to how the Y chromosome is passed on in humans. However, the social chromosome in red fire ants appeared less than one million years ago, making it much younger than the human Y chromosome, which is 180 million years old. This makes the social chromosome a good candidate for examining the early evolution of special chromosome variants that are only inherited.

How differences between the SB and the Sb chromosomes are evolving is an open question, however. Perhaps each version of the social chromosome has been optimised through natural selection to one colony type. Another suggestion is that the Sb chromosome has degenerated over time because its genes cannot be 'reshuffled' as they would be on normal chromosomes.

Martinez-Ruiz et al. compared genetic variants on the SB and Sb chromosomes, along with their expression in different types of ant colonies. The analysis showed that the Sb variant is in fact breaking down because of the lack of gene shuffling. This loss is compensated by intact copies of the same genes found on the SB variant, which explains why ants with the Sb variant can only survive if they also carry the SB version. Only a handful of genes on the social chromosomes appear to have been optimised by natural selection. Therefore Martinez-Ruiz et al. concluded the differences between the two chromosomes that lead to different colony types are collateral effects of Sb's inability to reshuffle its genes.

This work reveals how a special chromosome similar to the Y chromosome in humans evolved. It also shows how multiple complex evolutionary forces can shape a species' genetic makeup and social forms.

recombination rather than by antagonistic selection (*Branco et al., 2018*; *Cavoto et al., 2018*; *Branco et al., 2017*; *Ma et al., 2020*).

The social supergene of the red fire ant *Solenopsis invicta* is an excellent model for comparing how evolutionary antagonism and the consequences of suppressed recombination shape early supergene evolution. This species includes single-queen and multiple-queen colonies, which differ in behavior, physiology and ecology. This social polymorphism is controlled by a pair of 'social chromosomes', SB and Sb, that carry distinct supergene variants (*Wang et al., 2013*). In single-queen colonies, workers and queens are SB/SB homozygotes. In multiple-queen colonies, egg-laying queens are SB/Sb heterozygotes, but workers can either be homozygous or heterozygous. Sb/Sb queens are rare and their offspring are unviable (*Gotzek and Ross, 2007*). Furthermore, because of at least three inversions (*Stolle et al., 2019*; *Huang et al., 2018*), recombination is severely repressed between SB and Sb (*Wang et al., 2013*; *Pracana et al., 2017a*). Similar to a Y chromosome, Sb thus lacks recombination opportunities. Importantly, the fire ant supergene system could be as young as 175,000 generations. It thus offers a rare glimpse into the evolutionary forces shaping the early stages of supergene evolution (*Wang et al., 2013*).

Suppressed recombination reduces the efficacy of purifying selection through Hill-Robertson interference, which can lead to gene degeneration (*Charlesworth, 2016*). Accordingly, Sb has accumulated non-synonymous single nucleotide substitutions and repetitive elements (*Stolle et al., 2019*; *Pracana et al., 2017a*). Despite Sb degeneration, gene content in SB and Sb is highly similar (*Wang et al., 2013*; *Pracana et al., 2017a*), likely because of the system's young age and purifying

selection in haploid males (*Hall and Goodisman, 2012*). Modifications in the genomic sequence can lead to gene expression changes (*Denver et al., 2005*; *Rifkin et al., 2005*), the accumulation of mutations in Sb alleles of some genes could therefore result in SB/Sb expression bias. Changes in expression due to gene degeneration should result in specific expression patterns. For instance, if such mutations go to fixation because of inefficient purifying selection, they would likely result in consistent allelic bias across all tissues. Alternatively, if such mutations go to fixation because they are adaptive responses to antagonistic selection, their expression levels would be likely to require tissue-specific fine-tuning for particular functions. This process should result in tissue-specific allelic bias, as observed across human tissues (*GTEx Consortium, 2015*).

The accumulation of deleterious mutations in Sb alleles should lead to lower expression levels due to impacts in regulatory sequences. The opposite effect seems to be less likely and may only involve highly specific mutations (*Loewe and Hill, 2010*). Additionally, selection against the expression of deleterious alleles should result in the downregulation of some Sb alleles (*Ma et al., 2020*; *Xu et al., 2019*; *Pucholt et al., 2017*). If lower Sb expression occurs for dosage-sensitive genes, selection should favor upregulation of the corresponding SB alleles. Such dosage compensation can occur through different mechanisms. For instance, in male *Drosophila melanogaster* the entire X chromosome is expressed at roughly twice the level of autosomes to compensate for the lack of expression in the highly degenerated Y chromosome (*Conrad and Akhtar, 2012*). In species with much younger sex chromosomes such as *Drosophila miranda*, dosage compensation instead occurs only at a gene-by-gene level (*Alekseyenko et al., 2013*; *Nozawa et al., 2018*). Because the fire ant supergene is young, we hypothesize that individual genes for which the Sb allele has begun to degenerate would show evidence of dosage compensation. For such genes, the SB allele would be more highly expressed than the Sb allele, while their combined total expression should be similar between SB/Sb and SB/SB individuals.

Because the supergene variants determine social forms in the fire ant, their evolution will also have been shaped by antagonistic selection. We thus expect that the supergene region is enriched for genes with alleles that are beneficial for one of the social forms but detrimental for the other (*Zemp et al., 2016*; *Vicoso et al., 2013*; *Khil et al., 2004*; *Parsch and Ellegren, 2013*). Genes with such alleles are more likely to also show differences in expression between social forms, resulting in an enrichment of socially biased expression in the supergene. However, such enrichment could instead emerge from lowered expression associated with gene degeneration (*Ma et al., 2020*), or the fixation of mutations with neutral phenotypic effects (*Harrison et al., 2012*). Adaptation through lower expression of Sb alleles is indistinguishable from gene degeneration. In contrast, increased expression of Sb alleles is more likely to result from adaptation (*Harrison et al., 2012*; *Pál et al., 2001*). This line of reasoning has been used in the analysis of sex chromosomes, in which patterns of high sexually-biased expression for sex-chromosome linked genes is used as a proxy for benefit (*Mank, 2017*; *Mank et al., 2013*; *Zhou and Bachtrog, 2012*). Following this logic and given the expected general pattern of Sb downregulation due to degeneration, genes with high expression of the Sb allele are more likely to be adaptive. Furthermore, because Sb should be enriched in alleles beneficial for multiple-queen colonies, we expect that genes with high expression in multiple-queen colonies will tend to show Sb bias.

We disentangle these evolutionary processes by generating detailed genomic and transcriptomic data: we sequenced genomes of fire ants from their native South American range and combined these with existing genomes from the invasive North American range. This enables us to identify genes with fixed differences between the SB and Sb variants of the social chromosome. To detect differences in expression between SB and Sb alleles, we performed RNA sequencing (RNA-seq) from SB/Sb individuals from the South American range of the species. We used three body parts of queens (head, thorax and abdomen) and whole bodies of workers. We combined this data with published RNA-seq data from SB/Sb individuals from invasive North American and Taiwanese populations. We then compared these expression patterns with those obtained from comparisons between social forms.

We find that most genes in the social chromosome show no strong allelic bias and that there is no clear pattern of supergene-wide expression bias towards either variant. Genes with biased expression tend to show patterns consistent with gene degeneration, such as lower Sb expression with increasing numbers of non-synonymous mutations. Additionally, we find that more genes are SB biased than expected given differences in expression between social forms, a pattern consistent

with partial dosage compensation. Accordingly, the accumulation of non-synonymous mutations in Sb alleles correlates with an allelic bias towards SB, consistent with ongoing Sb degeneration. Despite these observations, we find an overrepresentation of Sb-biased genes among genes with higher expression in individuals from multiple-queen colonies. This result indicates that antagonistic selection has also shaped the expression patterns in the fire ant supergene. Given the observed impact of gene degeneration, our results highlight the importance of considering the genomic context of gene expression patterns before making inferences about functionality.

## Results

### Hundreds of genes have fixed allelic differences between supergene variants

To identify differences between supergene variants, we obtained 408-fold genome coverage from 20 haploid SB males and 20 haploid Sb males. Thirteen within each group (65%) were from the native South American range whereas the rest were from an invasive North American population (*Wang et al., 2013*). By comparing the two groups of males we identified 2877 single nucleotide polymorphisms (SNPs) with one allele in all SB individuals and a different allele in all Sb individuals, affecting 352 genes (*Supplementary file 1*). Among the 3.4% of SNPs affecting coding sequence, almost half changed the amino-acid sequence, with one change to a premature stop codon (47.7% non-synonymous *vs.* 52.3% synonymous changes). The remaining SNPs were in intergenic (36.1%), intronic (58.0%) or in untranslated regions (2.5%).

Because the invasive North American population went through a severe bottleneck in the 20th century (*Ascunce et al., 2011*), we repeated the analysis after separating populations. We found 252 additional SNPs with fixed differences between SB and Sb individuals in South America, and 23,022 additional fixed differences between SB and Sb in North America. The latter number is 4-fold higher than expected due to differences in sampling size alone and is in line with lower genetic diversity of both supergene variants in North America due to the invasion bottleneck.

### Seven genes have consistent variant-specific expression patterns in all populations

To understand the impacts of different evolutionary processes on the supergene, we compared the expression of SB alleles and Sb alleles for all genes in the region. For this, we generated RNA-seq data from whole bodies of SB/Sb workers and from abdomens, thoraces, and heads of SB/Sb virgin queens collected in South America. To compare with patterns in other fire ant populations, we additionally incorporated existing RNA-seq gene expression data from pools of whole bodies of SB/Sb queens collected in the USA and Taiwan (*Wurm et al., 2011*; *Fontana et al., 2020*). We summarize our hypotheses and results in *Table 1*.

Among the 352 genes with fixed differences between SB and Sb in this combined dataset, 122 had sufficient expression for analysis of differences between alleles. We found that seven of the genes (5.7%) had consistent expression differences between variants across all populations (linear mixed-effects model; Benjamini-Hochberg (BH) adjusted p<0.05, *Figure 1*). Expression bias went in both directions: the Sb variants of 'pheromone-binding protein Gp-9/OBP3' (LOC105194481), 'retinol-binding protein pinta-like' (LOC105199327) and uncharacterized LOC105193135 were consistently more highly expressed. In contrast, the SB variants of 'ejaculatory bulb-specific protein 3' (LOC105199531), 'carbohydrate sulfotransferase 11-like' (LOC105193134), 'calcium-independent phospholipase A2-gamma' (LOC105203065) and uncharacterized LOC105199756 were consistently more highly expressed.

We repeated our analysis in a population-specific manner. Within each of our three populations, this approach independently confirmed most of the allelic biases we had previously detected (*Figure 1—figure supplement 1*). It additionally uncovered population-specific patterns of allelic bias within the supergene (*Figure 1—figure supplements 2–4*, *Supplementary file 3*).

**Table 1.** Summary of the hypotheses proposed in this study, the tests carried out to explore them, the data used and the results obtained.

| Underlying process | Hypothesis | Test | Expectation | Data | Result |
|---|---|---|---|---|---|
| Suppressed recombination | Allele bias is determined by the effects of genomic structure independently of function | Allele specific expression in the supergene remains constant across body parts/castes | Tissue-specific allele specific expression would suggest fine-tuning by selection for specific functions. A lack of allelic bias would be consistent with random changes in expression due to a random accumulation of mutations in Sb | RNAseq from three body parts of queens and whole bodies of workers from South American populations. Data generated in this study. | We find no tissue-specific differences in allelic bias (124 genes, four levels, DESeq2 Wald test > 0.05, *Figure 1*) |
| | | Allele specific expression differences within the supergene should be highly correlated between closely related populations | Highly correlated supergene expression patterns between closely related populations would indicate that most expression differences between variants depend on the genomic content, rather than on function. Alternatively, if most expression patterns are driven by function, gene expression differences between supergene variants should be consistent across different populations, irrespective of ancestry. | RNAseq from SBSb queens from populations in the invasive range of North America and Taiwan and the native range of South America. North American and Taiwanese populations are closely related. Data generated in this study and from *Wurm et al., 2011* and *Fontana et al., 2020*. | North American and Taiwanese expression patterns within the supergene are highly correlated ($r^2$=0.67). Correlation between South American and the invasive populations is lower ($r^2$=0.21 and 0.18 for North America-South American and Taiwan-South America respectively). *Figure 1—figure supplement 5*. |
| | Some genes will show signs of dosage compensation | Genes Sb alleles with a hallmark of past sequence degeneration are more likely to be biased towards SB | A positive correlation between the number of non-synonymous mutations in Sb and lower allelic expression would indicate gene degeneration. | RNAseq from SBSb queens from populations in the invasive range of North America and Taiwan and the native range of South America. Data generated in this study and from *Wurm et al., 2011* and *Fontana et al., 2020*. | SB bias increases with the number of non-synonymous mutations in Sb. *Figure 4—figure supplement 2* |
| | | We should find genes with a strong allelic bias towards the SB allele, but with no expression differences between social forms | In some genes, deleterious mutations in Sb leading to lowered expression of this allele should result in an increased expression of the SB allele to reach balanced expression, resulting in similar expression levels between multiple-queen (SB/Sb) and single-queen (SB/SB) individuals. Alternatively, low expression of the Sb allele will invariably lead to low expression of multiple-queen individuals. | RNAseq from North American SBSb and SBSB queens. Data from *Wurm et al., 2011* and *Fontana et al., 2020*. | The patterns of expression differences between social forms in the social chromosome cannot be explained only by the observed allelic biases. A model allowing for dosage compensation fits the data best (294 genes, *Figure 3*). Differences in gene expression between social forms does not vary with varying levels of allelic bias (193 genes, *Figure 3—figure supplement 1*). Most genes with a strong SB bias are not differentially expressed between SBSb and SBSB queens (12 out of 15, binomial test p=0.03, *Figure 4c*) |

*Table 1 continued on next page*

*Table 1 continued*

| Underlying process | Hypothesis | Test | Expectation | Data | Result |
|---|---|---|---|---|---|
| Antagonistic selection | Selection favors the linkage of antagonistic alleles to the supergene | The supergene region should be enriched in genes with differences between social forms | Selection would favor the linkage to the supergene of genes with different expression optima for the different social forms. This would result in more genes with socially biased expression than expected by chance in the supergene. If selection does not play a major role in supergene expression patterns, socially biased genes will be equally spread throughout the genome. | RNAseq from North American SBSb and SBSB queens. Data from *Wurm et al., 2011* and *Fontana et al., 2020*. | The supergene region contains more genes with expression differences between social forms than expected by chance (33 out of 474 in the supergene, 260 out of 10,007 in the rest of the genome, Chi$^2$ test $p<10^{-7}$, *Figure 2a*). |
| | Selection favors the fixation of alleles adapted to the multiple-queen form in the Sb variant | The Sb variant is enriched in genes high multiple-queen expression | If selection has favored the linkage to Sb of alleles beneficial to multiple-queen individuals, this variant should be enriched in genes highly expressed in this social form despite widespread degeneration. Alternatively, most Sb alleles are expected to be downregulated due to degeneration, and consequently Sb should be enriched in genes with low expression in multiple-queen colonies. | RNAseq from North American SBSb and SBSB queens. Data from *Wurm et al., 2011* and *Fontana et al., 2020*. | Genes with Sb bias tend to show higher multiple-queen expression (5 out of 8, compared with 1 out of 15 for SB, Chi$^2$ test $p=0.02$, *Figure 4d*). This pattern is unlikely to be neutral due to widespread gene degeneration (*Figure 3*) and given that social bias patterns are similar within the supergene and in the rest of the genome (29 out of 33 genes with multiple queen bias in the supergene compared with 245 out of 260 in the rest of the genome, Chi$^2$ test $p=0.31$, *Figure 2b*). |

## Supergene allele expression differs based on population ancestry but not body parts

If allelic bias evolved in response to antagonistic selection, we expect it to be fine-tuned for specific functions across different tissues. In contrast, if allelic bias resulted from the accumulation of neutral or deleterious mutations, we would expect a consistent bias across tissues.

To test which scenario occurred in the fire ant, we used the South American RNA-seq data. We found no effect of body part or caste on allelic expression for any gene in the supergene region (DESeq2's Logarithmic Ratio Test and all pairwise Wald comparisons between interaction terms; all BH adjusted $p>0.05$). This result was unlikely to be due to lack of power because we did find such differences for genes in normally recombining parts of the genome, despite having less power to do so (see Materials and methods).

The general lack of tissue-specific fine-tuning of allelic expression bias in the supergene suggests that most of this bias is due to the accumulation of neutral or deleterious mutations (*Stolle et al., 2019*; *Pracana et al., 2017a*) in Sb rather than being adaptive. We further tested this idea by comparing expression patterns between populations. If most changes in Sb were not adaptive, patterns of allelic bias should correlate with similarity of the supergene genomic sequence. We therefore expected a stronger positive correlation of allelic bias between the two closely related invasive populations than to the less closely related South American population (*Ascunce et al., 2011*). For each pair of populations, we calculated correlations of the log2 ratios between the expression levels of SB and Sb alleles (*Figure 1—figure supplement 5*). Our findings were in line with our expectation: correlation was stronger between invasive populations (Spearman's $r^2 = 0.67$), than between either invasive and the native population (Spearman's $r^2 = 0.21$ and 0.18). This result was further supported by a linear mixed-effects model of all data: population ancestry has a stronger effect on allelic

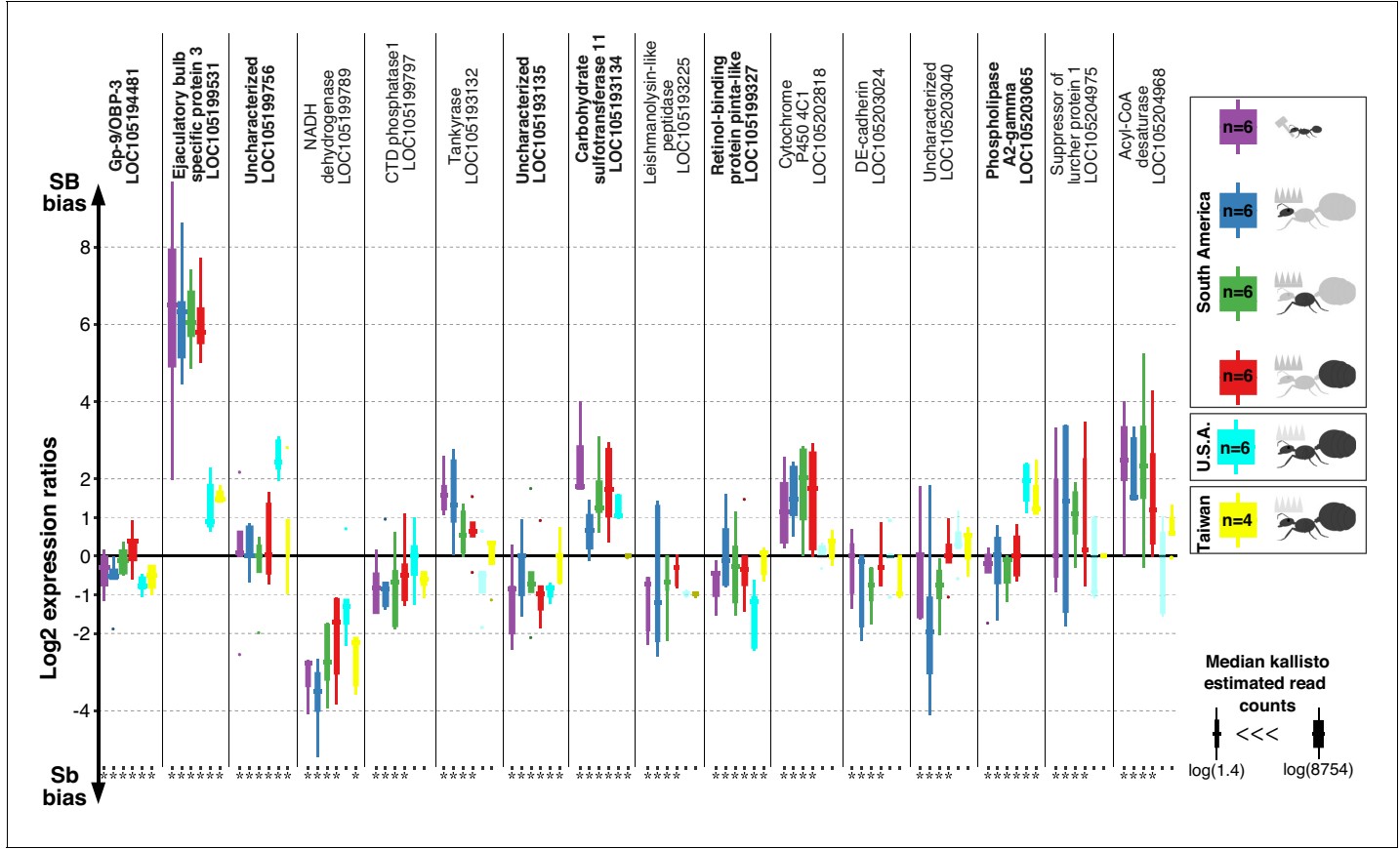

**Figure 1.** Differences in expression levels between alleles for genes in the fire ant social supergene in heterozygous SB/Sb individuals which exist only in multiple-queen colonies. Differences in expression (y axis) between social chromosome variants in whole bodies of workers from South America, heads, thoraces and abdomens of queens from South America, whole bodies of queens from North America and Taiwan. We show all 16 genes with significant allelic bias in South American populations, and the corresponding biases from the other populations. Bold gene names highlight when allelic bias occurs in all populations (Benjamini-Hochberg (BH) adjusted p<0.05 from the joint linear mixed-effects model). Significance in population-specific comparisons is indicated by an asterisks under each boxplot (BH adjusted p<0.05 from either the joint analysis or DESeq2 Wald tests). Each box shows the distribution of logarithm two expression ratios between SB and Sb in each sample type. Positive values indicate higher SB expression; negative values indicate higher Sb expression. A log2 expression ratio greater than one or smaller than −1 represents a two-fold gene expression difference in either direction. Genes are in chromosomal order.

The online version of this article includes the following source data and figure supplement(s) for figure 1:

**Source data 1.** Differences in expression levels between alleles for genes in the fire ant social supergene in heterozygous SB/Sb individuals which exist only in multiple-queen colonies.

**Figure supplement 1.** Overlapping number of genes with allele-specific expression according to comparisons in each population independently or after combining data from all populations.

**Figure supplement 2.** Allele-specific expression for genes in the fire ant social supergene for South American samples (information from body parts of queens and whole bodies of workers merged together).

**Figure supplement 2—source data 1.** Allele-specific expression for genes in the fire ant social supergene for South American samples.

**Figure supplement 3.** Allele-specific expression for genes in the fire ant social supergene for whole bodies of North American queens.

**Figure supplement 3—source data 1.** Allele-specific expression for genes in the fire ant social supergene for whole bodies of North American queens.

**Figure supplement 4.** Allele-specific expression for genes in the fire ant social supergene) for whole bodies of Taiwanese queens.

**Figure supplement 4—source data 1.** Allele-specific expression for genes in the fire ant social supergene) for whole bodies of Taiwanese queens.

**Figure supplement 5.** Correlation of log2 allele-specific expression ratios between the SB and Sb variants in heterozygous queens from three populations: South American data we generated, North American data (from *Morandin et al., 2016*), and Taiwanese data (from *Fontana et al., 2020*).

expression bias than geographic proximity (respective interaction terms with the effect of gene: F = 3.42, p<10$^{-15}$ and F = 0.94, p=0.65).

Together, these results support the idea that the effects of suppressed recombination on genomic architecture explain most of the allelic biases in the supergene region.

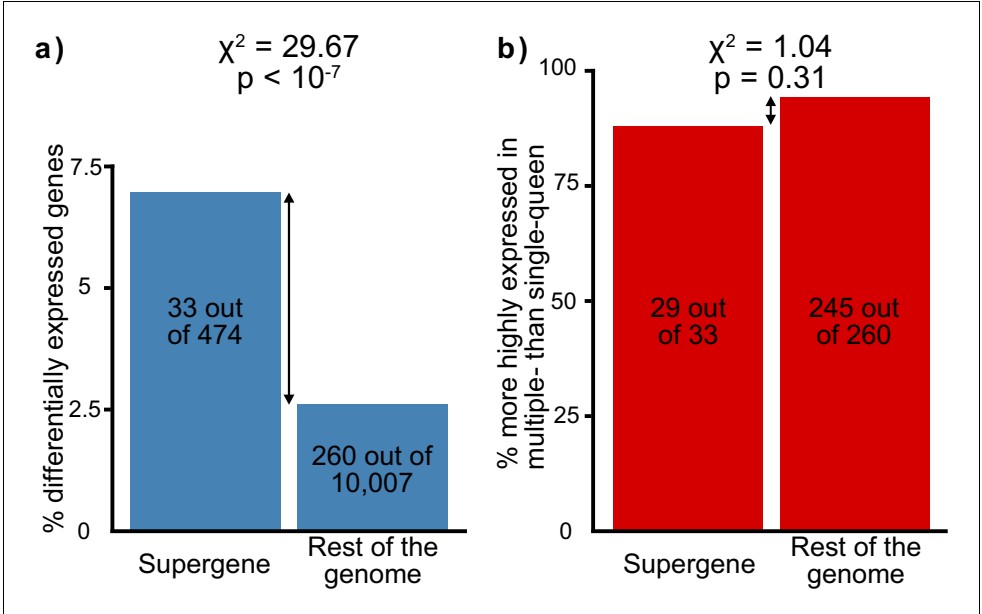

**Figure 2.** Distribution of socially biased genes in the genome of the red fire ant within (left bars) and outside (right bars) the supergene region. (a) The supergene region is significantly enriched in genes with differences between social forms, a pattern that could indicate the effect of antagonistic selection. (b) Most genes with differential expression between social forms are more highly expressed in multiple-queen colonies. This expression bias is observed across the genome and is not a unique feature of the supergene.

The online version of this article includes the following source data for figure 2:

**Source data 1.** Genomic location of the analyzed genes in the Solenopsis invicta genome.

## Overrepresentation of socially biased genes in the social supergene

Social-form specific selection should lead to an overrepresentation of socially biased gene expression in the supergene region. To test whether this pattern occurs, we compared gene expression between egg-laying queens from single-queen and from multiple-queen colonies. There were 293 socially biased genes with known chromosomal locations (*Supplementary file 4*). Such genes were indeed overrepresented in the supergene region (*Figure 2a*, 33 out of 474, 12 expected by chance, $\chi^2 = 29.7$, p<$10^{-7}$). Next, we examined the direction of expression bias: we found that most socially biased genes had higher expression in multiple-queen colonies than in single-queen colonies (274 out of 293, *i.e.*, 94%; binomial test, p<$10^{-15}$). However, this pattern was not specific to the supergene ($\chi^2$=1.04, p=0.3, *Figure 2b*). In sum, more socially biased genes are present in the supergene than in the rest of the genome, but the direction of social bias is similar across the genome. Since the trend of social bias is genome-wide, it cannot be explained by Sb degeneration alone.

## Genes with no social bias tend to have allele-specific expression biased towards the SB variant

Gene degeneration in Sb could lead to dosage compensation. To test whether this occurs, we compared the differences in expression levels between the SB and Sb alleles within heterozygous SB/Sb individuals from multiple-queen colonies to differences in expression between queens from single-queen (SB/SB) and multiple-queen colonies (SB/Sb). Dosage compensation should lead to a pattern where higher expression of the SB allele does not result in differences in expression between social forms.

We tested whether such a pattern occurs for 294 genes in the supergene region using North American data. We compared the proportion of SB allele expression ($P_B$) for each gene, with its expression in multiple-queen colonies relative to single-queen colonies ($P_{MQ}$; *Figure 3*). If Sb degeneration had led to its global downregulation or inactivation, we would have found a stark bias towards SB, irrespective of social bias. Instead, we find that the expression levels of SB and Sb are

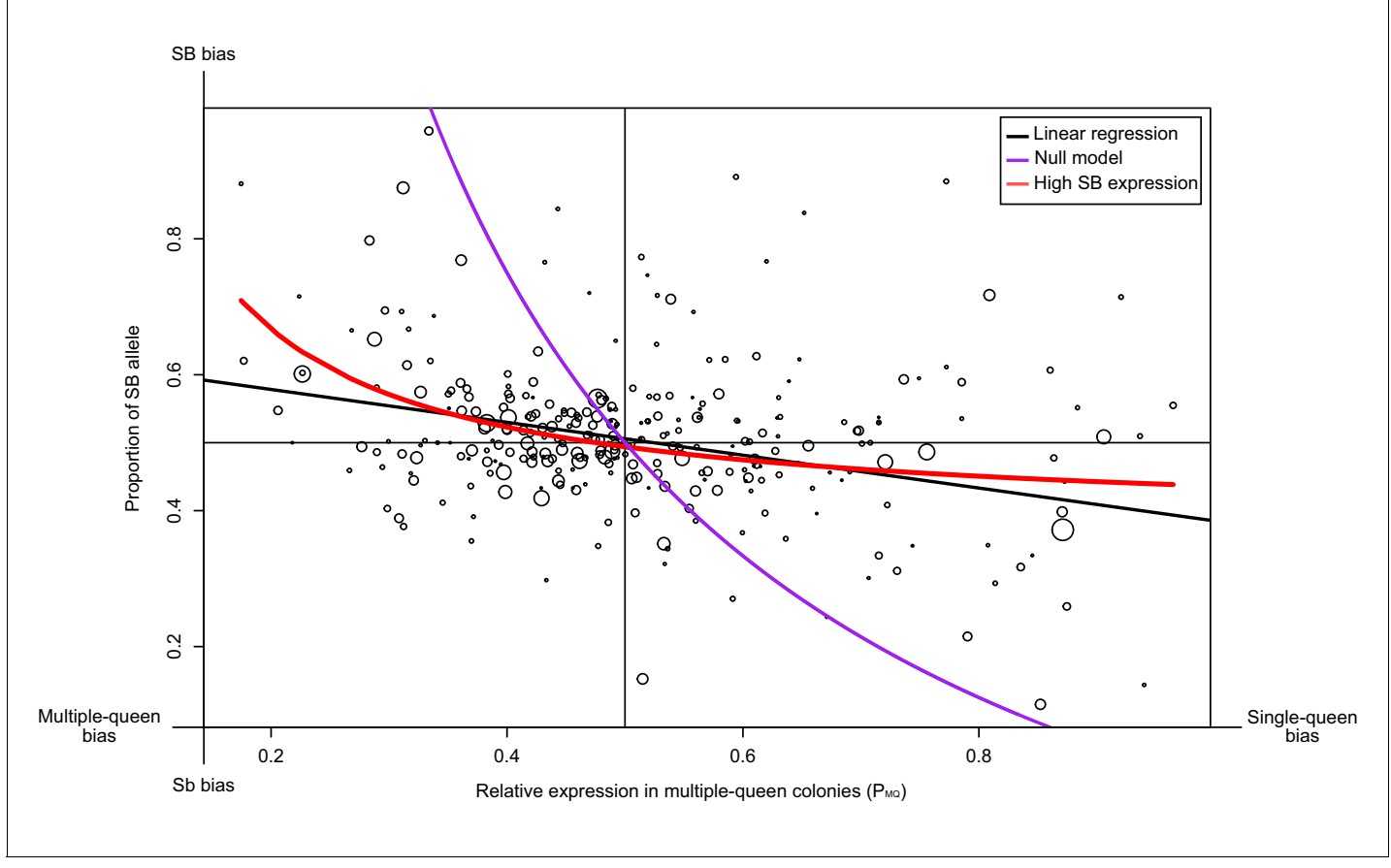

**Figure 3.** Relationship between measures of bias in allelic expression ($P_B$) and between social forms ($P_{MQ}$). Each point is one of 294 genes within the supergene (North American data). Point size is proportional to the mean expression in queens from multiple-queen colonies. The values were calculated as $P_B = x_B/(x_{B} + x_b)$ and $P_{MQ} = x_{MQ}/(x_{SQ} + x_{MQ})$; where $x_B$ and $x_b$ are the expression of SB and Sb alleles, and $x_{SQ}$ and $x_{MQ}$ are the expression in single-queen and multiple-queen colonies. Values of $P_{MQ}$ below 0.5 therefore indicate higher expression in SB/SB queens from single-queen colonies; values above 0.5 indicate higher expression in SB/Sb queens from multiple-queen colonies. Values of $P_B$ above 0.5 indicate allelic bias towards SB; values below 0.5 indicate bias towards Sb. The straight black line shows a linear regression. The purple line shows the predicted null relationship if the pattern of expression was due to Sb degeneration alone $P_{MQ} = (1/(2P_B+1))$. This model is a poor predictor of the data. The red line assumes gene-specific dosage compensation, where a decrease in expression of Sb leads to increased SB expression $P_B=(1-(P_{MQ}/2))/P_{MQ}$. The model including dosage compensation fits the data better than the null model; both models are significantly different (analysis of variance between models p<$10^{-5}$). The observed enrichment of multiple-queen genes in Sb is therefore unlikely due to Sb degeneration alone.

The online version of this article includes the following source data and figure supplement(s) for figure 3:

**Source data 1.** Social and allelic bias for genes in the supergene.

**Figure supplement 1.** Gene expression levels in single-queen and multiple-queen individuals for different levels of SB and Sb expression levels in multiple-queens.

**Figure supplement 1—source data 1.** Median gene expression levels in single-queen and multiple-queen individuals for different levels of SB and Sb expression levels in multiple-queens.

balanced for most genes without differences between social forms (black line in *Figure 3*, linear regression passes through the point 0.5, 0.5 with a non-significant deviation of 0.0058; p=0.32). At the extremes of the distribution, however, allelic bias is higher when social bias is higher. We therefore considered two models to explain this pattern. The null model assumes that differences in expression patterns between social forms are due solely to differences in baseline allelic expression (purple line in *Figure 3*). In this model, the $P_B$ and $P_{MQ}$ values change solely as a consequence of the relative expression of the Sb and the SB alleles (see Materials and methods). This model (the purple line) fits the data poorly. The second model (red line), additionally allows for the effect of gene-specific dosage compensation by increasing expression of B-alleles in SB/Sb individuals in multiple queen-colonies. This model fits the data much better and it is significantly different from the null

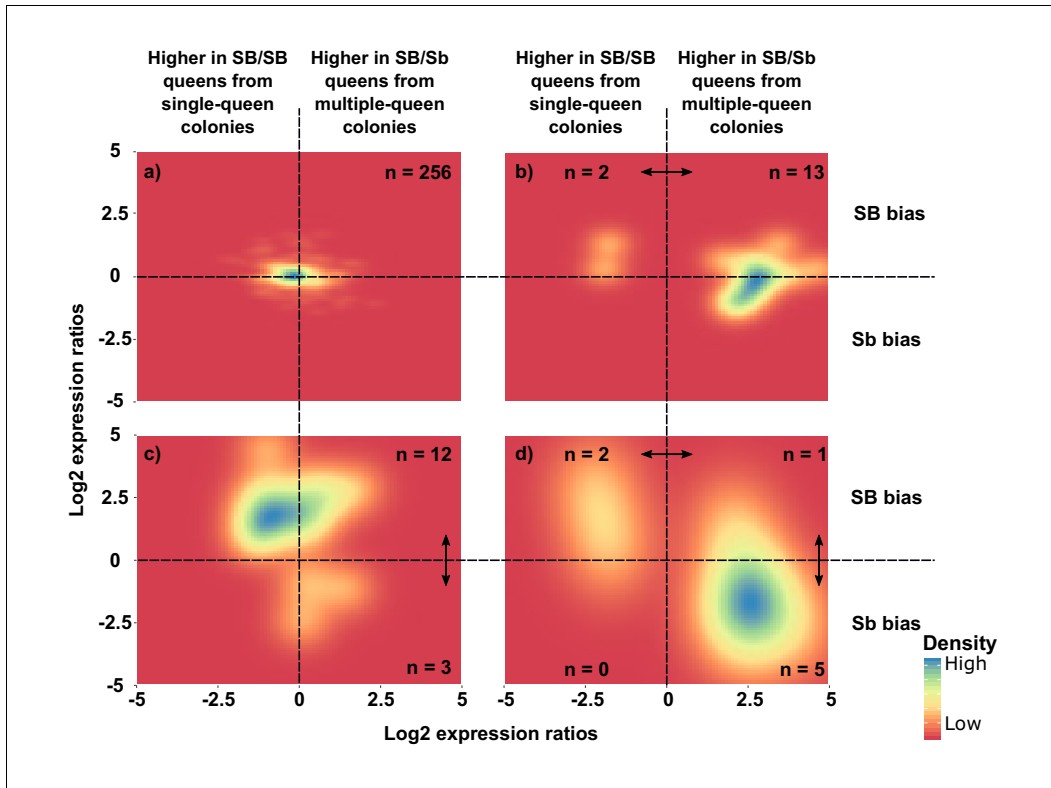

**Figure 4.** Distribution of differences in gene expression between social forms and between supergene alleles. X axes indicate ratios of expression between SB/Sb queens and SB/SB queens. Y axes indicate allelic expression ratios in SB/Sb queens. Both ratios use a log2 scale whereby log2 = 0 indicates absence of differences. Colors are proportional to numbers of genes. Double-headed arrows indicate significant expression differences. Panel (**a**) shows expression patterns for genes showing no difference in either comparison. The remaining three panels summarize expression patterns for: (**b**) genes with significant expression differences between SB/Sb and SB/SB queens only – these are biased towards higher expression in multiple-queen colonies (13 multiple-queen *vs.* two single-queen, binomial test, p=0.007); (**c**) genes with significant expression differences only between SB and Sb alleles within SB/Sb individuals – these are biased towards higher expression in the SB variant, in line with a dosage compensation mechanism (12 SB *vs.* 3 Sb, binomial test, p=0.03); (**d**) genes with significant expression differences between SB/Sb and SB/SB queens and between the SB and Sb variants in SB/Sb queens – the genes with higher expression of the Sb allele (y < 0) tend to be more highly expressed in queens from multiple-queen colonies (x > 0), in line with evolutionary antagonism between social forms (5 out of 8 Sb biased genes with bias towards multiple-queen colonies, compared with 1 out of 15 for SB biased genes $\chi^2$ = 5.8, p=0.02). The numbers in a) indicate how many genes had no differential expression. In b), (**c**) and d) the numbers in each quadrant indicate how many genes were significantly differentially expressed in the relevant comparison.

The online version of this article includes the following source data and figure supplement(s) for figure 4:

**Source data 1.** Expression differences between allele expression levels and between social forms for genes in the supergene.

**Figure supplement 1.** Overlap of genes with expression differences between variants and social forms out of all genes within the supergene region with sufficient data in both comparisons in the North American dataset.

**Figure supplement 2.** Allelic bias measured as the log2 ratio of expression between the SB and Sb alleles compared to the number of nonsynonymous mutations per gene.

**Figure supplement 2—source data 1.** Effect of non-synonymous mutations in the allelic bias between SB and Sb.

model (analysis of variance p<$10^{-5}$). This finding confirms that SB expression in SB/Sb individuals is higher than expected given the observed patterns of expression differences between social forms. Further in line with this, we found no relation between the relative expression levels of the SB and Sb alleles in queens from multiple queen-colonies to the total expression of the genes in the supergene in either social form (*Figure 3—figure supplement 1*;Wilcoxon sum rank test p>0.05). Thus,

higher SB allele expression in heterozygous individuals does not imply higher expression in single-queen colonies.

To complement these analyses of general trends, we focused on individual genes. Most genes had no significant allelic or socially biased expression (256, *i.e.*, 87%; *Figure 4a* and *Figure 4—figure supplement 1*). However, fifteen of the 294 genes showed allelic bias but no differences in gene expression between social forms (5%; *Figure 4c*). Most of these genes (12) had higher expression in SB, with only three being more highly expressed in Sb (binomial test, p=0.03). In line with this result, the median expression of SB alleles of these 15 genes was 5.9-fold higher than that of Sb alleles (Wilcoxon signed rank test p=0.0008 against equal expression).

The general and gene-specific patterns are hallmarks of dosage compensation: Lower expression of Sb alleles is compensated for by higher expression of SB alleles, resulting in the absence of expression differences between social forms. Importantly, the overall trends and gene-specific patterns both indicate that this dosage compensation occurs in gene-specific manners rather than being a response to global Sb-inactivation.

Dosage compensation would arise if gene degeneration leads to decreased Sb expression. We tested this hypothesis by asking whether genes with higher coding sequence degeneration have a higher SB bias. We used the number of non-synonymous mutations per gene in the Sb allele as a proxy for gene degeneration and compared this measure against allelic bias (*Figure 4—figure supplement 2*). Indeed, as the number of non-synonymous mutations increases, so does the allelic bias towards SB (coefficient = 0.052, p=0.01). This indicates that coding-sequence degeneration could lead to lower expression, or alternatively that some genes generally degenerate faster than others.

### Genes with higher expression of Sb than SB alleles are socially biased towards higher expression in multiple-queen colonies

Antagonistic selection should lead to an enrichment of genes in the supergene that are highly expressed in multiple-queen colonies and show allelic bias towards Sb. In contrast, without antagonistic selection all expression differences between Sb and SB alleles should be due to gene degeneration, and lead to lower Sb expression levels (*Ma et al., 2020*).

From the 294 genes analyzed in the previous section, eight (3%) had both allele-biased and socially-biased expression (*Figure 4d*). Their expression patterns were strongly directionally biased towards higher expression in Sb and in multiple-queen colonies (5 out 8 Sb biased genes were significantly more highly expressed in multiple-queen colonies *Figure 4d*; compared to 1 out of 15 for SB biased genes, $\chi^2$ = 5.8, p=0.02; *Figure 4c*). This enrichment of multiple-queen biased genes in the Sb variant is consistent with antagonistic selection. Although unlikely, Sb-specific upregulation could lead to higher expression in multiple-queen colonies without affecting fitness. However, the broad expression patterns described in the previous section (*Figure 3*) show that differences in expression between social forms are not due exclusively to changes in expression levels between Sb and SB alleles. We additionally showed that the fixation of non-synonymous mutations in Sb alleles correlates with lower expression levels (*Figure 4—figure supplement 2*). We therefore consider it unlikely that the trend of enrichment in high multiple-queen expression among Sb biased genes would have arisen neutrally. Consistent with the lack of direct correlation between allelic bias and social bias, only 3 of the seven genes with consistent allelic bias in all populations were also differentially expressed between social forms (*Supplementary file 2*): 'pheromone-binding protein Gp-9' (LOC105194481), 'ejaculatory bulb-specific protein 3' (LOC105199531) and 'retinol-binding protein pinta-like' (LOC105199327). This narrow overlap between allelic and social bias makes these genes candidates for playing roles in phenotypic differences between social forms.

## Discussion

In the fire ant, a supergene system with two variants (SB and Sb) controls whether colonies have one or multiple queens. We compared gene expression patterns between the SB and Sb variants of the social supergene within heterozygote SB/Sb individuals which exist only in multiple-queen colonies. We contrasted these patterns with differences in expression between queens from single-queen and multiple-queen colonies. We find patterns consistent with degeneration of Sb and with dosage compensation in response to this degeneration. We also find that some genes in Sb have patterns consistent with evolutionary antagonism.

## The effects of gene degeneration on supergene expression patterns

We found that a small proportion of the genes in the supergene region showed consistent allele-specific expression differences between the SB and Sb variants. It is tempting to conclude that such gene expression differences arose through selection, as a consequence of evolutionary antagonism between the single-queen and multiple-queen phenotypes. However, this interpretation may be too simplistic, as it ignores the impacts of supergene degeneration. Several studies have shown that Sb is degenerating (*Wang et al., 2013*; *Stolle et al., 2019*; *Pracana et al., 2017a*). Our observation that roughly half of the fixed differences in coding sequence between SB and Sb impact the protein sequence (where they are likely to have a deleterious effect) is also consistent with degeneration of the Sb variant. Such sequence-level degeneration is a symptom of reduced selection efficacy. By examining gene expression, we revealed three additional symptoms of degeneration and low selection efficacy. First, the absence of tissue-specificity in our study would not be expected if expression differences were adaptive. For some genes, the expression differences between Sb and SB are likely due to mutations that are completely neutral or deleterious. However, for other genes, expression differences could be partially adaptive, but low selection efficacy may have hindered the fine-tuning of their expression during the short timespan since the supergene's emergence (*Wang et al., 2013*). As a result, strong selection for a particular level of allele-specific expression in one body part (*e.g.*, in antennae), could result in consistent allele-specific expression patterns across tissues, even if this has mildly deleterious effects (*e.g.*, in the gut).

Second, there was a strong correlation in allelic bias between the invasive North American and Taiwanese populations, despite the data from the two populations being from different studies. Both invasive populations have lower genetic diversity overall (*Ascunce et al., 2011*), and in the supergene region in particular (*Pracana et al., 2017a*). The strong effect of ancestry on allelic bias indicates that genomic architecture, rather than gene function, defines most expression patterns within the supergene. Finally, gene degeneration can result in lower expression levels (*Xu et al., 2019*; *Pucholt et al., 2017*). For example, gene expression is reduced in genes with sequence-level signatures of degeneration in the mating-type chromosomes of the anther-smut fungus *Microbotryum* (*Ma et al., 2020*). We similarly find that Sb alleles with more non-synonymous mutations tend to have lower expression than SB alleles. This pattern could result from the relative inefficacy of selection on Sb, leading to selection favoring the downregulation of alleles accumulating detrimental mutations. We find such patterns of degeneration despite the likely effects of antagonistic selection (*Huang and Wang, 2014*) and the lack of evolutionary strata (*Pracana et al., 2017a*), both of which are likely to dampen the degeneration signal (*Ma et al., 2020*).

## Dosage compensation in the social supergene

The reduction in expression of the Sb allele in several genes could result in detrimental fitness effects if the genes involved are dosage sensitive, as observed in many sex chromosomes (*Mank, 2013*). We tested this idea in the fire ant supergene and found that allelic expression bias is relatively balanced, with similar levels of SB and Sb bias across the supergene. However, far more genes have multiple-queen biased expression than single-queen biased expression. This pattern implies that much of the observed SB bias leads to no differences between social forms. Our findings are consistent with dosage compensation, where the higher SB expression effectively counteracts lower Sb expression.

Some of this dosage compensation likely emerged through selection despite Hill-Robertson effects and a short time of divergence (*Pracana et al., 2017a*). However, some dosage compensation could instead occur automatically, whereby transcriptional machinery is less able to bind to degenerate regions of Sb and thus binds to SB instead (*Teufel et al., 2019*). Additionally, we speculate that regulatory elements located outside the supergene region could have co-evolved with Sb degeneration, allowing for variant-specific expression regulation. These elements would not be affected by suppressed recombination, allowing for a quicker emergence of dosage controlling mechanism (*Lenormand et al., 2020*).

Regardless of the mechanisms which mediate gene-by-gene dosage compensation in the fire ant, we show that it can emerge over time scales as short as approximately 175,000 generations (1 million years) (*Wang et al., 2013*; *Pracana et al., 2017a*). Furthermore, this is to our knowledge only the second known instance of dosage compensation in a supergene that does not determine sex or

mating type. Indeed, a 2–3 million-year-old supergene controlling color morphs of the white-throated sparrow also shows patterns consistent with dosage compensation (*Sun et al., 2018*). Such findings support the idea that many of the patterns seen in sex chromosomes are representative of supergenes in general. Indeed, rapid evolution of dosage compensation similarly occurred in the 10 million-year-old sex chromosomes of the plant *Silene latifolia* (*Muyle et al., 2012*) and in two *Drosophila* species neo-sex chromosomes that are only a few million years old (*Nozawa et al., 2014*; *Alekseyenko et al., 2013*; *Nozawa et al., 2018*).

## Supergene expression patterns consistent with antagonistic selection

Most Sb-biased genes are also more highly expressed in multiple-queen colonies, supporting the idea that antagonistic alleles are present in the supergene. In the unlikely case that this pattern had no fitness effects, it could arise neutrally (*Harrison et al., 2012*). We argue against this possibility because, as discussed above, expression patterns in the supergene do not follow what would be expected if differences between social forms were due exclusively to differences in allelic expression. Additionally, the patterns of expression differences between social forms are similar within the supergene and in the rest of the genome. This observation further suggests that expression differences between social forms are driven by factors unrelated to the genomic architecture of the supergene. Given the patterns of ongoing degeneration in Sb, we conclude that genes with Sb and multiple-queen bias were likely under antagonistic selection. Similarly, the neo-Y chromosome of *Drosophila miranda* is enriched in male-biased genes in the gonads (*Zhou and Bachtrog, 2012*).

## Candidate genes for differences between social forms

The approaches underpinning our analyses are unable to detect allelic differences in genes absent from the reference genome such as OBP-Z5, a putative Odorant Binding Protein exclusive to Sb (*Pracana et al., 2017b*). However, our analysis does single out candidate genes that potentially contribute to the social polymorphism of the fire ant (lists of all sequence and expression differences are in *Supplementary files 1*, *2*, *3*, *4*). In particular, three genes stood out because they were differentially expressed between social forms and had variant-specific allele expression in all populations. For the first gene, 'Pheromone-binding protein Gp-9' (LOC105194481), also known as OBP-3, the Sb allele was more highly expressed. For decades, this gene has been a candidate effector for social form differences (*Pracana et al., 2017a*; *Keller and Ross, 1998*), yet its linkage to hundreds of other genes in the supergene led to doubts that its association to social form is any more than coincidental. We found five fixed differences between SB and Sb for this gene, four of which could affect protein efficiency (consistent with previous findings [*Krieger and Ross, 2002*]). For the second gene, 'Ejaculatory bulb-specific protein 3' (LOC105199531), which also contains an insect odorant binding protein domain (InterPro IPR005055), the SB allele was more highly expressed. Orthologs of this gene are associated with mating (*Laturney and Billeter, 2014*) in *Drosophila melanogaster*, sexual behavior in a moth (*Bohbot et al., 1998*), subcaste differences in bumblebees (*Wolschin et al., 2012*), venom production in social hornets (*Yoon et al., 2015*) and caste differences in the termite *Reticulitermes flavipes* (*Steller et al., 2010*). Finally, LOC105199327 is likely a Pinta retinol-binding protein. Such proteins help pigment transport and vision in *D. melanogaster* and the butterfly *Papilio xuthus* (*Pelosi et al., 2018*). In sum, all three candidate genes have putative functions related to environmental perception, in line with the complex social phenotype requiring subtle changes in environmental perception or signaling (*Favreau et al., 2018*).

## Conclusions

We found differences in expression between the SB- and Sb-linked alleles of genes in the fire ant supergene across three populations. Such strong patterns can naively be assumed to be indicative of adaptive processes emerging from evolutionary antagonism between social forms. However, we show that the evolutionary forces shaping expression patterns in the supergene are complex and must be interpreted with care. In particular, genes with higher expression of the SB allele than the Sb allele tend to either lack expression differences between social forms or have lower expression in multiple-queen than in single-queen colonies. Both patterns are consistent with the idea that suppressed recombination leads to degeneration in Sb and thus lower Sb allele expression. In some cases, a dosage compensation mechanism through higher expression of the healthy SB allele leads

to similar expression levels in both social forms. In cases where no dosage compensation occurs, overall expression is lower in multiple-queen colonies than in single-queen colonies.

Conversely, we also show that genes with higher expression of the Sb allele than the SB allele are also biased towards higher expression in multiple-queen colonies. This pattern is consistent with evolutionary antagonism favoring the accumulation of beneficial alleles for multiple-queen individuals in Sb, the supergene variant found only in this social form.

Our study shows that multiple complex evolutionary forces can simultaneously act on a young supergene system. It highlights that allele-specific expression patterns alone are insufficient for inferring whether they are adaptive, deleterious or phenotypically neutral. Instead, putting such expression differences into broader contexts is needed to draw reasonable conclusions. Applying this idea to our data highlights genes which have molecular roles that could affect perception or signaling of the social environment.

# Materials and methods

## Key resources table

| Reagent type (species) or resource | Designation | Source or reference | Identifiers | Additional information |
|---|---|---|---|---|
| Biological sample (*Solenopsis invicta*) | Fire ants | Argentina | NCBI taxonomy 13686 | |
| Chemical compound, drug | Tri Reagent | Sigma-Aldrich | MFCD00213058 | DNA/RNA extraction |
| Commercial assay, kit | NEBNext Ultra II RNA Library Prep Kit | New England Biolabs | E7775L | |
| Commercial assay, kit | NEB Library Quant Kit | New England Biolabs | E7630 | |

## RNA sequencing of fire ants

We used three published and one new RNA-seq gene expression datasets from fire ants. *Wurm et al., 2011* obtained whole-body RNA-seq data from six pools of 4 egg-laying SB/Sb queens, each from a multiple-queen colony from Georgia, USA. *Morandin et al., 2016* obtained whole-body RNA-seq data from six samples, each being a pool of 3 queens from a single-queen or a multiple-queen colony (three replicates per social form) from Texas, USA. All the queens were mature and egg-laying (C. Morandin, personal communication), thus queens from multiple-queen colonies carried the SB/Sb genotype (*Keller and Ross, 1998*). Additionally, we manually checked the resulting RNA reads for heterozygous positions at key supergene markers using the IGV gene browser v2.4.19 (*Robinson et al., 2011*). *Fontana et al., 2020* generated RNA-seq data from 4 samples of SB/Sb queens from multiple-queen colonies from Taiwan. Each sample is a pool of whole bodies from two virgin queens (more details in *Supplementary file 5*).

All three published datasets are from pools of whole bodies from the invasive ranges of *S. invicta.* The Taiwanese invasive population of red fire ants is derived from that of North America (*Ascunce et al., 2011*). Because comparisons of whole bodies can be confounded by allometric differences (*Johnson et al., 2013*) and genetic diversity is reduced among Sb haplotypes in the invasive populations (*Pracana et al., 2017a*), we generated a new gene expression dataset. We collected samples from the native South American range of this species to obtain molecular data ((collection and exportation permit numbers 007/15, 282/2016, 433/02101-0014449-4 and 25253/16). To obtain RNAseq, we collected six multiple-queen colonies. We confirmed the social form of each colony (*Krieger and Ross, 2002*) on a pool of DNA from 10 randomly chosen workers. Colonies spent six weeks under semi-controlled conditions before sampling (natural light, room temperature, cricket, mealworm and honey water diet). From each colony, we snap-froze one worker and one unmated queen for gene expression analysis between 12:00 and 15:00 local time. To partly control for allometric differences between genotypes, we separated each queen into head, thorax and abdomen. This was done in petri dishes over dry ice using bleached tweezers. In total, we had 24 samples for RNA extraction: six whole bodies of workers and six replicates of three body parts from queens (more details in *Supplementary file 5*).

We extracted RNA and DNA from each sample using a dual DNA/RNA Tri Reagent based protocol (https://www.protocols.io/view/rna-dna-extraction-protocol-bi8fkhtn). We applied the *Krieger and Ross, 2002* assay on the extracted DNA to identify only individuals with the SB/Sb genotype. Once RNA was extracted, we prepared Illumina sequencing libraries from total RNA using half volumes of the NEBNext Ultra II RNA Library Prep Kit. We checked RNA and library qualities on an Agilent Tapestation 2200; library insert size averaged 350 bp. An equimolar pool of the 24 libraries was sequenced on a single lane of Illumina HiSeq 4000 using 150 bp paired-end reads. This produced an average of 14,848,226 read pairs per sample (maximum: 27,766,980; minimum: 6,015,662. Raw RNA-seq reads for all samples are on NCBI SRA (PRJNA542606).

For all datasets, we assessed read quality using fastQC (v0.11.5; http://www.bioinformatics.babraham.ac.uk/projects/fastqc/). Raw reads for all samples were of sufficient quality to be used in subsequent analysis. We removed low quality bases using fqtrim with default parameters (v0.9.5; http://ccb.jhu.edu/software/fqtrim/), and Illumina adapters using Cutadapt v1.13 (*Martin, 2011*). We then generated a STAR v2.5.3a (*Dobin et al., 2013*) index of the *S. invicta* reference genome (version gnG; RefSeq GCF_000188075.1 [*Wurm et al., 2011*] while providing geneset v000188075.1 in GFF format through the 'sjdbGTFtagExonParentTranscript = Parent' option. As recommended by the developers of STAR, we aligned each sample to the reference twice, using the 'out.tab' file for the second run, and set 'sjdbOverhang' to the maximum trimmed read length minus one, that is 74 for the *Wurm et al., 2011* data and *Morandin et al., 2016* data, 125 for the *Fontana et al., 2020* data and 149 for the South American data we generated here. Alignments were run using GNU Parallel v20150922 (*Tange, 2011*). All steps and downstream analyses were performed on the Queen Mary University of London's Apocrita High Performance Computing Cluster (*King et al., 2017*).

We further assessed aligned reads (*i.e.*, BAM files) using MultiQC v1.5 (*Ewels et al., 2016*) and the BodyGene_coverage.py script of the RSeQC toolkit v2.6.4 (*Wang et al., 2016*). We removed one sample from multiple-queen colonies in the Morandin et al. data from subsequent analyses due to poor alignment quality. None of the other BAM files showed markers of technical artefacts that could bias our results.

## Identifying SNPs with fixed differences between SB and Sb males

To detect allele-specific differences between SB and Sb we first identified SNPs with fixed differences between the SB and Sb variants. Because the patterns of genetic diversity differ between the invasive and South American *S. invicta* populations (*Ross et al., 2007*; *Ahrens et al., 2005*), we estimated allele specific expression differences in the social chromosome independently for each population. For this we used haploid male ants because they can provide unambiguous genotypes. For the invasive populations, we identified fixed allelic differences between a group of 7 SB males and a group of 7 Sb males from North America (NCBI SRP017317) (*Wang et al., 2013*).

For the South American population, we sequenced the genomes of 13 SB males and 13 Sb males sampled from across Argentina. For each individual, we extracted 1 µg of genomic DNA using a phenol-chloroform protocol. The extracted material was sheared to 350 bp fragments using a Covaris (M220). We constructed individually barcoded libraries using the Illumina TruSeq PCR-free kit. The libraries were quantified through qPCR (NEB library quant kit). An equimolar pool of the 26 libraries was sequenced on a HiSeq4000 at 150 bp paired reads. This produced an average of 17,790,416 pairs of reads per sample, with a maximum of 38,823,285 and a minimum of 7,910,042 (*Supplementary file 6*, genomic reads of all samples deposited on NCBI SRA (PRJNA542606)). For each dataset, we identified fixed allelic differences between the group of SB males and the group of Sb males. We first aligned the reads of each sample to the *S. invicta* reference genome (*Wurm et al., 2011*; gnG assembly; RefSeq GCF_000188075.1) using Bowtie2 v2.3.4 (*Langmead et al., 2009*). We then used Freebayes v1.1.0 (*Garrison and Marth, 2012*) to call variants across all individuals (parameters: ploidy = 1, min-alternate-count=1, min-alternate-fraction=0.2). We used BCFtools (*Li et al., 2009*) and VariantAnnotation (*Obenchain et al., 2014*) to only retain variant sites with single nucleotide polymorphisms (SNPs), with quality value Q greater than or equal to 25, and where all individuals had a minimum coverage of 1. To avoid considering SNPs erroneously called from repetitive regions that are collapsed in the reference genome, we discarded any SNP with mean coverage greater than 16 for the North American samples or 12 for the South American samples or where any individuals had less than 60% reads supporting the called allele. This last filtering step also acts to remove SNPs called from reads with sequencing errors. We

then extracted only the SNPs located within the supergene (based on the genomic locations from *Pracana et al., 2017a*) and with fixed differences between SB and Sb. This step was performed independently for each population. The two resulting variant call files were inspected using VCFtools v0.1.15 (*Danecek et al., 2011*) and we manually ensured that all variants had the SB allele as reference and Sb allele as alternative. To test the effect of sample size differences between populations we downsampled the South American dataset to 7 pairs of SB and Sb males, matching the sample size in the North American dataset.

We extracted SNPs shared between South and North American populations using BCFtools isec v1.9 (*Li et al., 2009*). We then used SNPeff (*Cingolani et al., 2012*) to characterize the effects of individual SNPs.

## Estimating read counts from alternate supergene variants in heterozygous individuals

Because the reference genome for *S. invicta* is based on an SB individual, read mapping could be biased towards the SB variant in heterozygous individuals, resulting in false positive detection of allelic bias (*Castel et al., 2015*). To overcome this potential artifact, we called BCFtools consensus v1.9 (*Li et al., 2009*) once using North American Sb males and once using South American Sb males. We then aligned the RNAseq reads from each sample to the regular reference genome (version gnG; RefSeq GCF_000188075.1) and also, independently, to the most relevant of the modified references. For the reads from the Taiwanese population, we used the Sb reference using North American SNPs. For the alignment we used STAR with the same parameters as described above. We merged the two resulting BAM files from each sample using SAMtools v1.9 (*Li et al., 2009*). We then used the 'rmdup' function from the WASP pipeline (*Soneson et al., 2015*) to generate reference-bias free alignment files. The resulting BAM files can be considered reference bias free alignments. We added a reading group ID to each reference-bias free BAM file using the 'AddOrReplaceReadGroups' tool from Picard (v 2.7.0-SNAPSHOT; http://broadinstitute.github.io/picard/). We then ran all BAM files through GATK's 'ASEReadCounter' v 3.6–0-g89b7209 (*Wright et al., 2017*) with default options to obtain read counts for each allele. We performed this step once on each population independently.

We then imported the resulting allele-specific SNP read counts per sample generated by GATK into R v3.4.4 (*R Development Core Team, 2017*). We used the R packages 'GenomicRanges' v1.26.4 (*Lawrence et al., 2013*) and 'GenomicFeatures' v1.26.3 (*Lawrence et al., 2013*) along with the NCBI protein-coding gene annotation for *S. invicta* to identify which SNPs are in which genes. We estimated the total expression level for a particular allele (*i.e.*, the SB or Sb variant for any given gene) as the median of all SNP-specific read counts per gene and per variant. For instance, consider a gene with three fixed SNPs between SB and Sb for which the SB variants have support from 12, 15 and 18 reads, and the Sb variants from 5, 8 and 6 reads. In this particular case, we would report that the SB variant for this gene has an expression level of 15 reads and the Sb variant, six reads. If instead of this approach, we randomly select one of the possible SNPs for every gene, we find qualitatively similar results to those reported.

Additionally, to test whether we would be able to detect allele-specific expression changes across body parts and castes in the South American data, we calculated allele-specific expression in the whole genome as a positive control. We used the VCF file containing all SNPs in the 26 males collected from South America. We retained only SNPs with expression data in all samples and a median of at least 1 X RNA coverage in each allele across all samples. After filtering, 1096 SNPs remained for which we were able to test for allele-specific expression. We performed an allele specific expression analysis throughout the whole genome using body part and caste information from South American populations. Unlike the analysis of genes in the supergene region, in the whole genome analysis we cannot ensure that every individual is heterozygous for all SNPs. Indeed, the average frequency in the population for all the alleles analyzed was 0.41 with a standard deviation of ±0.2. This implies that both alleles were not necessarily present in all samples. We therefore had far less power to detect allele-specific expression across body parts using data from the whole genome than using SNPs from the supergene region only. Despite this lack of power, we were able to detect significant (Wald test BH adjusted p<0.05) allele-specific expression changes across body parts of queens and workers in 15 SNPs. These significant SNPs were distributed across nine genomic scaffolds. The significant differences in allele-specific expression were between a queen body part and whole bodies of workers.

## Identifying expression differences between the SB and Sb variants of the supergene

We imported the estimated read counts generated by Kallisto into R using Tximport v1.2.0 (*Soneson et al., 2015*) and DESeq2 v1.14.1 (*Love et al., 2014*). For every sample, read counts for the SB alleles and for the Sb alleles come from the same sequencing library, thus standard normalization methods (*Dillies et al., 2013*) are not applicable. As recommended by the developers of DESeq2 (*Love, 2018*), we thus deactivated normalization by setting SizeFactors = 1. For the North American and Taiwanese datasets (*Wurm et al., 2011*; *Fontana et al., 2020*), we only considered genes expressed in all samples for downstream analyses, whereas for the South American populations RNA dataset, we only analyzed genes expressed in all replicates of at least one body part.

To have the strongest possible analysis of expression between the SB and Sb variants of the supergene region, we performed a joint analysis of RNAseq data from Taiwanese, South and North American populations. The South American dataset includes body part information, which is absent in the North American dataset. We applied a linear mixed effects model on the log2 of the expression ratios between SB and Sb across populations and body parts, using the R packages lme4 v1.1–18.1 (*Bates et al., 2014*) and lmerTest v3.1–0 (*Kuznetsova et al., 2017*). We fitted the log2 expression ratios using a 0 intercept with gene, population and their interaction as fixed effects, and the interaction between gene and body part as random effects (formula: log2 expression ratio ~0 + gene * population + (1|body part:gene)).

We also performed an additional linear mixed-effects model to test the effect of geography and ancestry on the allele-specific expression patterns within the supergene. We grouped the populations by geographical proximity (North and South America *vs.* Taiwan) or by phylogenetic proximity (Taiwan and North America *vs.* South America). We then fitted the log2 expression ratios using a 0 intercept, the main effects of gene, ancestry and geographic proximity and the interactions between ancestry and gene and between geographic proximity and gene as fixed effects. As random effects we used again the interaction between gene and body part (formula: log2 expression ratio ~0 + gene * ancestry + gene * geography + (1|body_part:gene)). We then performed an analysis of variance on the model to estimate the size effects of each term.

For both models, the log2 expression ratios were weighed by a function of the total read counts per gene to reduce the impacts of genes with low expression which have extremely high variance. Here we only report the results of the fixed effects per gene after adjustment of p values for multiple testing following the Benjamini-Hochberg approach (*Benjamini and Hochberg, 1995*). For this joint analysis we only used genes that had fixed differences between SB and Sb in all three populations.

We additionally analyzed the allele-specific expression patterns between the SB and Sb variants of the supergene in each population independently using DESeq2 (*Love et al., 2014* following *Castel et al., 2015*. The model formula used for the South American RNA-seq data used 'body part' and 'colony of origin' as blocking factors, and allele-specific expression, that is 'variant effect', as variable of interest. This analysis allowed us to detect differences in expression between variants specific to body part. Preliminary analyses showed that the interaction between 'variant effect' and 'body part' had no significant effect in any of the genes, and consequently, only the main 'variant effect' was considered as the factor of interest for this analysis. The model formula for both the *Wurm et al., 2011* and the *Fontana et al., 2020* RNA-seq datasets included only whole bodies of queens. We thus used 'sample' as a blocking factor and 'variant effect' as variable of interest.

In all analyses, we report gene differences between variants as log2 expression ratios between the SB and the Sb counts. That is, genes with expression biased towards SB will produce positive log2 expression ratios whereas those biased towards Sb will produce a negative value. To check whether there was an overall bias towards either variant, we tested the significance of the deviation from 0 for the median log2 expression ratios between SB and Sb via a Wilcoxon sum rank test.

## Expression differences between single-queen and multiple-queen colonies

We determined the expression levels for all samples from the North American populations (*Morandin et al., 2016*) by using the count mode in Kallisto v0.44.0 (*Bray et al., 2016*) using *S. invicta* coding sequences. We imported the estimated counts into DESeq2 v1.14.1 (*Love et al., 2014*) using Tximport v1.2.0 (*Soneson et al., 2015*). We compared the DESeq2 normalized

expression levels between social forms, determining significance of differential expression using the default Wald test for pairwise comparisons between genes. We estimated the proportion of significantly differentially to non-differentially expressed genes within and outside the supergene region based on supergene region coordinates from *Pracana et al., 2017a*. We then used the R packages GenomicRanges and GenomicFeatures (*Lawrence et al., 2013*) along with the annotations of *S. invicta* coding sequences to locate each gene with expression information in the genome. Our analyses are restricted to the 10,481 known *S. invicta* genes that can be reliably placed within or outside the supergene region; other genes are on scaffolds which lack chromosomal locations (*Pracana et al., 2017a*).

## Expression differences between variants and social forms

We fitted a model to test whether there is a significant relationship between allele-specific expression differences between supergene variants in the *Wurm et al., 2011* dataset, and gene expression differences between social forms (log2 expression ratios using the *Morandin et al., 2016* dataset). We examined the overall trend in allele-specific expression patterns within the supergene (*i.e.*, any bias towards expression of either the SB or Sb allelic variant).

We obtained relative expression levels using DESeq2 for both comparisons: single-queen vs. multiple-queen expression for each gene ($X_{SQ}$ vs. $X_{MQ}$) from the *Morandin et al., 2016* dataset and expression of the SB allelic variant vs. the Sb ($X_B$ vs. $X_b$) within each gene from the *Wurm et al., 2011* dataset DESeq2 returned an estimate of $\log2(X_B/X_b)$ for the differences in expression between alleles and $\log2(X_{SQ}/X_{MQ})$ for the differences among colony types. We first generated a null model in which the $P_B$ and $P_{MQ}$ values change solely as a consequence of the relative expression (r) of the Sb allele ($x_b$) and the SB alleles ($x_B$), such that $x_b = r\,x_B$, and therefore:

$$\mathbf{P}_B = \frac{\mathbf{X}_b}{\mathbf{X}_b + \mathbf{X}_B}$$

and

$$\mathbf{P}_{MQ} = \frac{\mathbf{X}_b + \mathbf{X}_B}{\mathbf{X}_b + 3\mathbf{X}_B}$$

Notice that in this case the minimum value of $P_{MQ}$ = ⅓ would occur when there was no expression of the b allele ($x_b$ = 0). For greater values of $x_b$, we can solve the pair of equations to obtain the relationship:

$$\mathbf{P}_{MQ} = \frac{1}{2\mathbf{P}_B + 1}$$

The second model additionally allows for the effect of dosage compensation by increasing expression of B-alleles in SB/Sb individuals in multiple queen-colonies. To do this, $P_B$ is defined by the expression differences between social forms such that:

$$\mathbf{P}_B = \frac{1 - \frac{\mathbf{P}_{MQ}}{2}}{\mathbf{P}_{MQ}}$$

Finally, we test whether the two models are significantly different using a standard analysis of variance test using the aov function in R. The linear regressions and statistical tests were performed in R v3.4.4 (*Lawrence et al., 2013*).

We also explored whether genes with low Sb allele expression had higher SB allele expression, resulting in similar expression between multiple-queen (SB/Sb genotype) and single-queen (SB/SB genotype) individuals. Such a pattern would be consistent with an ongoing process of dosage compensation. To do so, we excluded the nine genes with significant biases towards Sb and high SB-SB/Sb ratios (*i.e.*, SB variant more highly expressed in SB/Sb than SB/SB individuals), since they are more likely to have been subjected to antagonistic selection. We also excluded genes with fewer than three read counts mapping to either allele to remove more noisy estimates. The rest of all analyzed genes were then grouped by relative SB/Sb expression. We then compared the overall expression levels between these groups in multiple-queen and single-queen individuals.

## Data availability

We deposited the genomic and transcriptomic reads we generated from South American *Solenopsis invicta* on NCBI SRA (PRJNA542606). All analysis scripts used will be made available at https://github.com/wurmlab/2019-11-allelic_bias_in_fire_ant_supergene (copy archived at https://github.com/elifesciences-publications/2019-11-allelic_bias_in_fire_ant_supergene; *Martinez-Ruiz, 2020*).

## Acknowledgements

This research was possible thanks to the funding provided by the Natural Environment Research Council (NE/L00626X/1 and NERC EOS Cloud to YW; NE/L002485/1 to CM-R), Deutscher Akademischer Austauschdienst (DAAD) Postdoc Program (570704 83 to ES); European Commission Marie Curie Actions (PIEF-GA-2013–623713 to ES and YW); Biotechnology and Biological Sciences Research Council (BB/K004204/1 to YW). We thank Emiliano Boné for help with ant rearing, Claudia Castillo Carrillo for help during sample collections, Dr. Monika Struebig for preparing RNA-seq libraries, Phillip Howard, Dr. Chloe Economou and Martin Tran for their technical wet lab support, Wurm lab members and colleagues in the Department of Organismal Biology for valuable input and discussions, and the ITS Research Group and Adrian Lärkeryd at Queen Mary University of London for computational support and access to the Apocrita High Performance Computing facility (http://doi.org/10.5281/zenodo.438045) and NERC EOS Cloud.

## Additional information

### Funding

| Funder | Grant reference number | Author |
| --- | --- | --- |
| NERC | NE/L00626X/1 | Yannick Wurm |
| NERC | NE/L002485/1 | Carlos Martinez-Ruiz |
| DAAD | 570704 83 | Yannick Wurm |
| European Commission Marie Curie Actions | PIEF-GA-2013-623713 | Yannick Wurm |
| BBSRC | BB/K004204/1 | Yannick Wurm |

The funders had no role in study design, data collection and interpretation, or the decision to submit the work for publication.

### Author contributions

Carlos Martinez-Ruiz, Conceptualization, Data curation, Formal analysis, Funding acquisition, Validation, Investigation, Visualization, Methodology, Writing - original draft, Writing - review and editing; Rodrigo Pracana, Data curation, Supervision, Validation; Eckart Stolle, Resources, Writing - review and editing; Carolina Ivon Paris, Resources; Richard A Nichols, Conceptualization, Formal analysis, Supervision, Validation, Investigation, Visualization, Methodology, Writing - review and editing; Yannick Wurm, Conceptualization, Resources, Data curation, Supervision, Funding acquisition, Investigation, Visualization, Methodology, Project administration, Writing - review and editing

### Author ORCIDs

Carlos Martinez-Ruiz  https://orcid.org/0000-0002-4817-0565
Yannick Wurm  https://orcid.org/0000-0002-3140-2809

### Ethics

Animal experimentation: We snap froze field-collected ants into liquid nitrogen. Ethical guidelines typically do not consider such invertebrates. However, we performed the experiments in a manner that minimized potential harm.

Decision letter and Author response
Decision letter https://doi.org/10.7554/eLife.55862.sa1
Author response https://doi.org/10.7554/eLife.55862.sa2

## Additional files

### Supplementary files

• Supplementary file 1. Single nucleotide polymorphisms (SNPs) with fixed differences between the SB and Sb variant in present in both North and South American populations of red fire ant. The columns show, from left to right: the scaffold in the reference genome of the fire ant (version gnG; RefSeq GCF_000188075.1) where the SNP is located, its position within the scaffold, the allele present in all SB males (reference allele), the allele present in all Sb males (alternative allele), the position within the gene were the SNP is located, the gene (or genes) that could be potentially affected by the SNP and the potential effect of the SNP in Sb: 'HIGH' implies a change that substantially alters protein sequence (e.g., an early stop codon), 'MODERATE' implies a change affecting protein sequence, but without necessarily altering substantially protein structure (e.g., a non-synonymous mutation), 'LOW' implies a change with no effect on protein sequence (e.g., a synonymous mutation) and 'MODIFIER' are variants outside gene coding regions that could have potential regulatory effects. The last three columns of the table are based on the results from snpEff.

• Supplementary file 2. Names and RefSeq identifiers of the seven genes that are significantly differentially expressed between the SB and Sb variants of the *S. invicta* supergene. The significance levels were determined using a linear mixed effect models on the log2 expression ratios between SB and Sb. Population was used as a random effect and the log2 expression ratios were weighted by read count of the gene. The third column in the table shows whether that particular gene is also differentially expressed in the comparison between social forms (using *Morandin et al., 2016* data), and if so, in which social form it is more highly expressed.

• Supplementary file 3. Genes with significant differential expression between the SB and Sb variants of the *S. invicta* supergene in a) South American, b) North American or c) Taiwanese populations. Significance levels were determined by the Wald test in DESeq2. Significance was established as Benjamini and Hochberg corrected p<0.05. The columns in the tables show the names of the genes, their RefSeq identifiers, their log2 expression ratios for allele-specific expression differences between variants (values greater than 0 are more highly expressed in SB) and in which variant they are more highly expressed.

• Supplementary file 4. Genes with significant differential expression between queens from single-queen and multiple-queen colonies of S. invicta from North American populations (data from *Morandin et al., 2016*). Significance levels were determined by the Wald test in DESeq2. Significance was established as Benjamini and Hochberg corrected p<0.05. The columns in the tables show the names of the genes, their RefSeq identifiers, their log2 expression ratios for gene expression differences between social forms (values greater than 0 are more highly expressed in queens from multiple-queen colonies) and in which social form they are more highly expressed. Locations in the supergene are based on the data from Pracana, Rodrigo, et al. Molecular ecology 26.11 (2017): 2864–2879.

• Supplementary file 5. Overview of RNA-seq datasets used in this study. (a) Accession numbers of the North American RNA-seq datasets. 'Project' and 'SRA' columns indicate NCBI identifiers. The descriptions provided and the sequencing method used are based on metadata available on NCBI and in the manuscripts. One sample (marked with an asterisk) was discarded because of very low coverage after aligning the reads to the *S. invicta* genome. b) Details for the South American RNA-seq dataset. From left to right, the colony name from where samples were taken, the caste used from these colonies, the body parts extracted, the location of each colony in Argentina, the coordinates from where the sample was taken and finally, whether or not samples from the same colony were used to generate the VCF with fixed differences between Sb and SB.

• Supplementary file 6. Location of the colonies used to estimate single nucleotide polymorphisms (SNPs) between SB and Sb males. Note that individuals from colonies AR102, AR111, AR112, AR114 and AR28 were also used to extract RNA for RNA sequencing.

• Transparent reporting form

## Data availability

We deposited genomic and transcriptomic reads we generated from South American Solenopsis invicta on NCBI SRA (PRJNA542606). All analysis scripts used are available at https://github.com/wurmlab/2019-11-allelic_bias_in_fire_ant_supergene (copy archived at https://github.com/elifesciences-publications/2019-11-allelic_bias_in_fire_ant_supergene).

The following dataset was generated:

| Author(s) | Year | Dataset title | Dataset URL | Database and Identifier |
|---|---|---|---|---|
| Martinez-Ruiz C, Pracana R, Stolle E, Paris CI, Nichols RA, Wurm Y | 2019 | Solenopsis invicta Raw sequence reads | https://www.ncbi.nlm.nih.gov/bioproject/PRJNA542606 | NCBI BioProject, PRJNA542606 |

The following previously published datasets were used:

| Author(s) | Year | Dataset title | Dataset URL | Database and Identifier |
|---|---|---|---|---|
| Wang J, Wurm Y, Nipitwattanaphon M, Riba-Grognuz O, Huang YC, Shoemaker D, Keller L | 2012 | Solenopsis invicta Variation | https://www.ncbi.nlm.nih.gov/bioproject/PRJNA182127 | NCBI BioProject, PRJNA182127 |
| Fontana S, Chang NC, Chang T, Lee CC, Dang VD, Wang J | 2019 | The fire ant social supergene is characterized by extensive gene and transposable element duplication | https://www.ncbi.nlm.nih.gov/bioproject/PRJNA542500 | NCBI BioProject, PRJNA542500 |
| Morandin C, Tin MM, Abril S, Gómez C, Pontieri L, Schiøtt M, Sundström L, Tsuji K, Pedersen JS, Helanterä H, Mikheyev AS | 2015 | Comparative transcriptomics reveals the conserved building blocks involved in parallel evolution of ant phenotypic traits | https://www.ncbi.nlm.nih.gov/bioproject/PRJDB4088 | NCBI BioProject, PRJDB4088 |
| Wurm Y, Wang J, Riba-Grognuz O, Corona M, Nygaard S, Hunt B, Ingram K, Falquet L, Nipitwattanaphon M, Gotzek D, Dijkstra M, Oettler J, Comtesse F, Cheng-Jen S, Wu W, Chin-Cheng Y, Thomas J, Beaudoing E, Pradervand S, Flegel V, Cook E, Fabbretti R, Stockinger H, Long L, Farmerie W, Oakey J, Boomsma J, Pamilo P, Yi S, Heinze J, Goodisman M, Farinelli L, Harshman K, Hulo N, Cerutti L, Xenarios I, Shoemaker D, Keller L | 2011 | The genome of the fire ant Solenopsis invicta | https://www.ncbi.nlm.nih.gov/bioproject/PRJNA49629 | NCBI BioProject, PRJNA49629 |

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
