## [Decision Letter]

**Acceptance summary:**

There is increasing evidence that supergenes play an important role in the evolution of diverse traits, including colour patterns, parasite resistance and, as in the current study, social interaction. We know very little about the early evolution of such supergenes. The study by Martinez-Ruiz et al. addresses this topic, studying the social supergenes of the red fire ant with the help of gene expressing experiments. The authors report a complex history of the evolution of this supergene including signatures of adaptive evolution, gene degeneration and gene-specific dosage compensation.

**Decision letter after peer review:**

Thank you for submitting your article "Genomic architecture and evolutionary conflict drive allele-specific expression in red fire ant's social supergene" for consideration by *eLife*. Your article has been reviewed by three peer reviewers, one of whom is a member of our Board of Reviewing Editors, and the evaluation has been overseen by Patricia Wittkopp (Senior Editor).

The reviewers have discussed the reviews with one another and the Reviewing Editor has drafted this decision to help you prepare a revised submission.

This is a rich paper with a lot of information, including various sources of information to create a complex picture. The authors use caste, social form, supergene genotype and geography to understand how variation in expression of genes forming the so-called social supergene can be used to unravel the evolution of an ant supergene alleles. This is a powerful and fruitful approach, but results in a complex and long manuscripts. Many of the comments of the reviewers refer to the style the material presented.

Furthermore, the reviewers ask for clarification of terms and concepts. Some assumptions are not well explained or justified.

Reviewer 4 suggests an overview summary table summary table synthesizing the main comparisons, results and conclusions and thus to guide the reader through the complex material. The text (in particular the Discussion) is at places long and windy. Here one may shorten the text and give it more of a red line. The summary table may help here. Below are the detailed comments of the three reviewers.

Reviewer #1:

In this study Martinez-Ruiz et al. investigate the social supergene in fire ants that has become famous as a "Green Beard gene" explaining the polymorphisms in queen numbers in colonies of the fire ant. The study uses a combination of population genomics and gene expression to characterize the differences among the two functional types of the gene. For this, the authors used a combination of different data sets (some new, some earlier published). Consistent with the relatively young age of this supergene, most variation in the super gene haplotypes is not associated with the phenotype. Also expected, the haplotypes suffer from degeneration, a phenomenon related to the absence of recombination and the increased likelihood of accumulation of deleterious mutations. Finally, a relatively small number of genes carry the signature of functional haplotypes, associated with the phenotypes. Interestingly, these signatures are consistent with the idea that the evolution of the supergene is driven by conflict. All in all, this is a solid study and I think it is of interest for many readers of *eLife*.

The study is well done and technically sound. The combination of different methods and different datasets places particular strength on the analysis. The results are appropriately discussed and placed in the light of other super genes known from a number of very different systems. I have no comments.

Reviewer #3:

This manuscript reports analyses on gene expression in the supergene controlling social structure in the red fire ant to explore whether differences in gene expression between the two haplotypes can be due to antagonistic selection or degeneration of the non-recombining haplotype. The questions are very interesting, the biological model is fascinating and the dataset is important. However, there is some room for improvement, in particular regarding clarity and interpretation, as detailed below.

– The Abstract is frustrating, it would be good to indicate what kind of evidence are provided in favor of each of the two hypotheses. Actually, the Introduction, and even the Results, are not clear either on the specific hypotheses tested and how they can be tested. It only becomes clear in the conclusion how the hypotheses were tested and how the results allow inferring the two types of selection. I would recomment to be clear in the Abstract and the end of the conclusion about what kinds of tests will be run and what results would support what hypothesis. Also in the Results section, help the reader to interpret the raw results. As it stands, the results read like a long list of statistical tests, which one has a hard time to make sense of.

– About the interpretation of degeneration causing some of the differences in gene expression, maybe it could be tested whether the alleles in the Sb alleles that are less expressed than in SB show more footprints of degenerative mutations, as was done in the following paper, that would be relevant to cite : Ma et al., 2020.

Reviewer #4:

This is a very rich paper with a complex analysis of gene expression, in which the authors use caste, social form, supergene genotype and geography to understand how variation in expression of genes forming the so-called social supergene can be used to unravel the evolution of supergene alleles. This is an interesting approach. This paper presents a large amount of data on gene expression, placed in an interesting context of supergene evolution and possibly allele degeneration. I like the idea that degeneration can be studied by exploring the possibility of dosage compensation.

I have two main points to raise regarding how the approach is explained and justified.

The first point is that the authors do not explain well enough why they seem to equate over-expression with adaptive variation in expression. It seems that in many places over-expression of genes in a particular compartment (body part, or cast, or genotype, etc.) is seen and presented as an indication that this gene is advantageous in that context. I can think of many genes whose overexpression is deleterious or lethal, and in many cases of alleles for which repressed expression is advantageous. It is also not clear why under- and over-expression should not be equally likely when it comes to mutations affecting regulatory regions, since transcription factors can also suppress gene expression; therefore, mutations that cannot be purged despite their deleterious effects could equally be mutations with a positive or negative effect on expression. The idea that an unpurged mutation is more likely to have a loss of function effect is also not so obvious, at least in the case of non-coding mutations. This idea that the direction of change in expression due to mutations indicates its adaptive value lingers in the manuscript, and while this may reflect a misinterpretation on my part, I suggest that the authors clearly explain, from the outset, what the expectations are as to what constitutes an expected adaptive response through variation in expression, and test it accordingly.

The second point is that the authors try to disentangle the effect of the social phenotype and supergene genotype, but the situation is not symmetrical because of the lethality of the Sb/Sb genotypes and the dominance of Sb. It may be expected that genes with an overexpression in an Sb genotype are only overexpressed in multiqueen colonies. Again, this may reflect my lack of knowledge of the system, but I think the authors should explain in more detail how they can eliminate the expected correlation between the Sb genotype and social form.

---

## [Author Response]

This is a rich paper with a lot of information, including various sources of information to create a complex picture. The authors use caste, social form, supergene genotype and geography to understand how variation in expression of genes forming the so-called social supergene can be used to unravel the evolution of an ant supergene alleles.This is a powerful and fruitful approach, but results in a complex and long manuscripts. Many of the comments of the reviewers refer to the style the material presented.

We are grateful for the constructive comments of the editors and reviewers. In light of these comments, we have re-written most of the Introduction and Discussion to make our message clearer and more concise. Due to these efforts to streamline the resulting manuscript, the main text is now also 6.4% shorter. We have paid particular attention to style, which we believe is now also much improved.

Furthermore, the reviewers ask for clarification of terms and concepts. Some assumptions are not well explained or justified.

We apologize for the ambiguities. We have now stated more explicitly our hypotheses in the Introduction and added more context to our results. We have additionally devoted more space to clearly defining our assumptions.

Reviewer 4 suggests an overview summary table summary table synthesizing the main comparisons, results and conclusions and thus to guide the reader through the complex material.

This is indeed a helpful idea. We have added Table 1, where we summarize in a concise manner all of our hypotheses, the tests carried out to address them, the expected results and the observed results.

The text (in particular the Discussion) is at places long and windy. Here one may shorten the text and give it more of a red line. The summary table may help here. Below are the detailed comments of the three reviewers.

Because of our efforts to make the text more readable and to the point the Discussion is now 27% shorter and the whole text is 6% shorter. Adding Table 1 indeed also helped contextualize our results more clearly. Overall, we believe that these efforts have substantially improved the quality of the manuscript.

Reviewer #3:This manuscript reports analyses on gene expression in the supergene controlling social structure in the red fire ant to explore whether differences in gene expression between the two haplotypes can be due to antagonistic selection or degeneration of the non-recombining haplotype. The questions are very interesting, the biological model is fascinating and the dataset is important. However, there is some room for improvement, in particular regarding clarity and interpretation, as detailed below.The Abstract is frustrating, it would be good to indicate what kind of evidence are provided in favor of each of the two hypotheses. Actually, the Introduction, and even the Results, are not clear either on the specific hypotheses tested and how they can be tested. It only becomes clear in the conclusion how the hypotheses were tested and how the results allow inferring the two types of selection. I would recomment to be clear in the Abstract and the end of the conclusion about what kinds of tests will be run and what results would support what hypothesis.

We agree that the message could have been clearer. We have now substantially revised the Abstract and Introduction. We have now explicitly stated the hypotheses we tested, how the tests were performed and how the conclusions we reached from these results. Additionally, we have added a summary of the hypotheses, tests and results in Table 1.

Also in the Results section, help the reader to interpret the raw results. As it stands, the results read like a long list of statistical tests, which one has a hard time to make sense of.

The Results section indeed previously included a lot of technical detail for several analyses that we had done as additional controls. This was in part due to merging what was initially in supplementary data into the main text. We now still mention most of these results but have substantially reduced the technical detail with which they are described. Furthermore, we now provide more context and interpretation within each subsection of Results, which should also help the reader make sense of the work.

About the interpretation of degeneration causing some of the differences in gene expression, maybe it could be tested whether the alleles in the Sb alleles that are less expressed than in SB show more footprints of degenerative mutations, as was done in the following paper, that would be relevant to cite : Ma et al., 2020.

These are indeed interesting ideas. We applied them to our system with some caveats. The system studied by Ma et al., 2020 determines mating types; it is thus not thought to be affected by evolutionary antagonism, whereas ours is. In addition, their system is older and contains evolutionary strata, unlike the fire ant supergene. Such differences could reduce potential signals. Furthermore, we lack genomes from sister species to test as many features as Ma et al. did. Despite all this, we were able to test whether there is a gene-level link between sequence degeneration and reduced expression. For this, we now compare allelic bias with the numbers of nonsynonymous mutations for each gene (in Figure 4—figure supplement 2). We additionally perform a linear regression to test whether increased gene degeneration in Sb leads to an overall SB bias. We find that genes with more nonsynonymous mutations are more biased towards SB expression, supporting Sb degeneration. In addition, we now compare changes in expression between social forms across a set of categories of SB/Sb allele bias (Figure 3—figure supplement 1). We find that relative expression between social forms remains constant across different levels of SB bias, adding further evidence for dosage compensation of SB. These results are described in subsection “Genes with no social bias tend to have allele-specific expression biased towards the SB variant”, and discussed in subsection “The effects of gene degeneration on supergene expression patterns”. We describe how these analyses were performed in subsection “Expression differences between variants and social forms”.

Reviewer #4:This is a very rich paper with a complex analysis of gene expression, in which the authors use caste, social form, supergene genotype and geography to understand how variation in expression of genes forming the so-called social supergene can be used to unravel the evolution of supergene alleles. This is an interesting approach. This paper presents a large amount of data on gene expression, placed in an interesting context of supergene evolution and possibly allele degeneration. I like the idea that degeneration can be studied by exploring the possibility of dosage compensation.I have two main points to raise regarding how the approach is explained and justified.The first point is that the authors do not explain well enough why they seem to equate over-expression with adaptive variation in expression. It seems that in many places over-expression of genes in a particular compartment (body part, or cast, or genotype, etc.) is seen and presented as an indication that this gene is advantageous in that context. I can think of many genes whose overexpression is deleterious or lethal, and in many cases of alleles for which repressed expression is advantageous. It is also not clear why under- and over-expression should not be equally likely when it comes to mutations affecting regulatory regions, since transcription factors can also suppress gene expression; therefore, mutations that cannot be purged despite their deleterious effects could equally be mutations with a positive or negative effect on expression. The idea that an unpurged mutation is more likely to have a loss of function effect is also not so obvious, at least in the case of non-coding mutations. This idea that the direction of change in expression due to mutations indicates its adaptive value lingers in the manuscript, and while this may reflect a misinterpretation on my part, I suggest that the authors clearly explain, from the outset, what the expectations are as to what constitutes an expected adaptive response through variation in expression, and test it accordingly.

The reviewer raises many valid points here that we hope to have addressed more succinctly throughout the text now.

Firstly, we would like to clarify that we do not expect selection to be completely suppressed in the supergene region. There is now indeed some evidence that it can occur at low frequency (Ross and Shoemaker, 2018). Hill-Robertson effects due to suppressed recombination make selection less efficient. Consequently, a broader range of mildly deleterious mutations would remain, but highly deleterious mutations will still be selected against. Bearing this clarification in mind, while we agree that some random mutations can increase expression, we disagree with the idea that over- and under- expression are equally likely. It is thought that most de novo mutations are mildly deleterious and therefore one can expect that mutations rather decrease expression due to impacting regulatory sequences, transcription factor binding sites or other functionally relevant sequences. The opposite effect seems to be less likely and may only involve mutations in specific regulatory (suppressor) sequences or need to involve very specific point mutations with transcription-increasing effects (Loewe and Hill, 2010). Genes with high expression generally show markers of adaptive selection such as slower evolution (e.g. Harrison, Wright and Mank, 2012; Pál, Papp and Hurst, 2001; Fraser, Moses and Schadt, 2010). In addition, previous studies show that genes in degenerating regions have lower expression levels (Xu et al., 2019; Pucholt et al., 2017; Ma et al., 2020).

Importantly, we would like to clarify that the patterns we refer to are general patterns – for multiple genes. We do not feel that expression patterns alone are sufficient do demonstrate that changes in any individual gene are adaptive or detrimental. What we mean instead is that, overall, we expect that groups of genes affected by gene degeneration will have general patterns of lower expression. Conversely, in general, highly expressed genes would be more likely to be adaptive than detrimental. We now describe these assumptions in more detail in the Introduction. Additionally, we have now added Figure 4—figure supplement 2, which shows with our data that genes with more nonsynonymous mutations in Sb are more likely to be biased towards SB, indicating that gene degeneration tends to result in lower Sb expression levels (as recommended above). Finally, we now discuss (subsection “Dosage compensation in the social supergene”) the potential mechanisms by which the phenotypic effects of gene degeneration could have arisen in such a young system.

The second point is that the authors try to disentangle the effect of the social phenotype and supergene genotype, but the situation is not symmetrical because of the lethality of the Sb/Sb genotypes and the dominance of Sb. It may be expected that genes with an overexpression in an Sb genotype are only overexpressed in multiqueen colonies. Again, this may reflect my lack of knowledge of the system, but I think the authors should explain in more detail how they can eliminate the expected correlation between the Sb genotype and social form.

Again, the reviewer raises a very good point. We have now re-written the Introduction and Discussion to focus on addressing the issue raised. The data and tests used for Figure 3 (previously Figure 4) show that the observed pattern of differences in expression between social forms are unlikely to be due solely to differences in expression levels between the SB and Sb alleles. We show that patterns of expression differences between social forms are better explained if we allow for an overall higher expression level of the SB alleles when Sb expression is low. Much of the observed SB bias results in no differences in expression between social forms. In other words, social bias cannot be directly linked to allelic bias in the supergene. In addition, in Figure 2B we show that the patterns of expression differences are similar between social forms within and outside the supergene. This result further supports the idea that patterns of social bias are more or less independent of allelic bias within the supergene.

Given these results, we conclude that there is not a direct link between allele expression level differences between SB and Sb and social bias in the supergene. We discuss these results in more detail in subsections “Overrepresentation of socially biased genes in the social supergene” and “Genes with no social bias tend to have allele-specific expression biased towards the SB variant” in the Results section and in “Supergene expression patterns consistent with antagonistic selection” in the Discussion section.

**References**

Fraser, Hunter B., Alan M. Moses, and Eric E. Schadt. 2010. “Evidence for Widespread Adaptive Evolution of Gene Expression in Budding Yeast.” Proceedings of the National Academy of Sciences of the United States of America 107 (7): 2977–82.